# Signature of spin-phonon coupling driven charge density wave in a kagome magnet

H. Miao [1] ✉, T. T. Zhang [2], H. X. Li [1,3], G. Fabbris [4], A. H. Said[4], R. Tartaglia[4,5], T. Yilmaz [6], E. Vescovo[6], J.-X. Yin[7], S. Murakami [2], X. L. Feng [8], K. Jiang [8], X. L. Wu[9], A. F. Wang [9] ✉, S. Okamoto [1] ✉, Y. L. Wang [10] ✉ & H. N. Lee [1]

The intertwining between spin, charge, and lattice degrees of freedom can give rise to unusual macroscopic quantum states, including high-temperature superconductivity and quantum anomalous Hall effects. Recently, a charge density wave (CDW) has been observed in the kagome antiferromagnet FeGe, indicative of possible intertwining physics. An outstanding question is that whether magnetic correlation is fundamental for the spontaneous spatial symmetry breaking orders. Here, utilizing elastic and high-resolution inelastic x-ray scattering, we observe a c-axis superlattice vector that coexists with the $2 \times 2 \times 1$ CDW vectors in the kagome plane. Most interestingly, between the magnetic and CDW transition temperatures, the phonon dynamical structure factor shows a giant phonon-energy hardening and a substantial phonon linewidth broadening near the c-axis wavevectors, both signaling the spin-phonon coupling. By first principles and model calculations, we show that both the static spin polarization and dynamic spin excitations intertwine with the phonon to drive the spatial symmetry breaking in FeGe.

The combination of magnetism and characteristic electronic structures of the kagome lattice, including flat-band[1–3], Dirac-fermion[4–7], and van Hove singularities[8,9], is a productive route to realize correlated and topological quantum states. Significant interests have been focused on a kagome superconductor $AV_3Sb_5$ ($A$ = K, Rb, Cs)[10], where van Hove singularities near the Fermi level trigger cascade time- and spatial-symmetry breaking orders[8–23]. Lately, a correlated version of $AV_3Sb_5$ is realized in a kagome magnet FeGe[24,25]. Like the $AV_3Sb_5$ ($A$ = K, Rb, Cs)[10], the electronic structure of FeGe features multiple van Hove singularities near the Fermi level, $E_F$[24,25]. A charge density wave (CDW) establishes in the A-type antiferromagnetic (A-AFM) phase and induces

physical consequences, including anomalous Hall effect[24] and robust edge modes[25], reminiscent to those observed in $AV_3Sb_5$[16,18]. Below the CDW transition temperature, $T_{CDW}$, the static spin polarization is enhanced, indicating an intimate correlation between spin, charge, and lattice degrees of freedom[24]. Despite these interesting observations, key questions yet to be answered. For instance, although CDW has been observed in correlated magnetic systems, such as the cuprate high-$T_c$ superconductors[26,27] and spin-density-wave systems[28], emergence of CDW well below the magnetic transition temperature is rare, suggesting a new correlation driven CDW mechanism in FeGe. Focusing on the kagome metals with van Hove singularities near $E_F$, it is also

[1]Materials Science and Technology Division, Oak Ridge National Laboratory, Oak Ridge, TN, USA. [2]Department of Physics, Tokyo Institute of Technology, Okayama, Meguro-ku, Tokyo, Japan. [3]Advanced Materials Thrust, The Hong Kong University of Science and Technology (Guangzhou), Guangzhou, China. [4]Advanced Photon Source, Argonne National Laboratory, Argonne, IL, USA. [5]"Gleb Wataghin" Institute of Physics, University of Campinas, Campinas, São Paulo, Brazil. [6]National Synchrotron Light Source II, Brookhaven National Laboratory, Upton, New York, USA. [7]Laboratory for Quantum Emergence, Department of Physics, Southern University of Science and Technology, Shenzhen, China. [8]Beijing National Laboratory for Condensed Matter Physics, and Institute of Physics, Chinese Academy of Sciences, Beijing, China. [9]Low Temperature Physics Laboratory, College of Physics and Center of Quantum Materials and Devices, Chongqing University, Chongqing, China. [10]School of Emerging Technology, University of Science and Technology of China, Hefei, Anhui, China. ✉e-mail: miaoh@ornl.gov; afwang@cqu.edu.cn; okapon@ornl.gov; yilinwang@ustc.edu.cn

urged to determine the geometry of the CDW in FeGe and its possible connection with the loop current scenario[11–17,24]. Here, we address these fundamental questions using advanced x-ray scattering and numerical calculations. We discover charge superlattice peaks at the A-AFM wavevectors in FeGe, differentiating the CDW geometry in FeGe and AV₃Sb₅ despite the same 2 × 2 × 2 superstructure[20]. Most interestingly, the phonon dynamical structure factor shows giant phonon hardening and large phonon broadening effect near the c-axis CDW wavevectors above the $T_{CDW}$. These phonon anomalies are in stark contrast with the phonon softening, known as Kohn anomaly, in the electron-phonon coupled systems and the emergent amplitude mode that hardens below the $T_{CDW}$[19]. Combining with density-functional theory (DFT) and model calculations, we show that the energetically favored 2 × 2 × 2 superstructure in FeGe involves mainly c-axis lattice distortions in the Kagome plane, which is stabilized by the strong spin-phonon interactions.

## Results and Discussion

FeGe adopts a hexagonal structure with space group P6/mmm (No. 191). It is composed of a kagome lattice of Fe atoms with Ge-1 centered in the hexagons. These kagome layers are stacked along the c-axis and separated by honeycomb layers of Ge-2. At $T_N = 410$ K, an A-AFM kicks in with spin moment pointing along the c-axis. Below $T_{CDW} \sim 110$ K, concomitant anomalous Hall effect and enhanced spin polarization are observed[24]. Fig. 1b shows the density functional theory plus dynamical mean field theory (DFT + DMFT) calculated spectral function of FeGe. In agreement with angle-resolved photoemission spectroscopy (ARPES) study[24], van Hove singularities at the **M** point (Fig. 1c) are pushed to the Fermi-level due to the local correlation effect (see also Supplementary Fig. 2). Fig. 1d, e show x-ray diffraction scans along high-symmetry directions at $T = 10$ K. Consistent with previous diffraction and scanning tunneling microscopy studies[24,25], CDW superlattice peaks are observed at $Q_M^{//}$ (H = 0.5, K,

L=integer) and $Q_L$ (H = 0.5, K L=integer+0.5), where H, K, L are reciprocal lattice directions as shown in Fig. 1c (see Supplementary Fig. 3). While superlattice peak positions at $Q_M^{//}$ and $Q_L$ are the same as AV₃Sb₅[19,20,22], as shown in Fig. 1e, we observe a new charge superlattice peak at $Q_A^{\perp} = (0, 0, 2.5)$ that is absent in AV₃Sb₅[19]. This new superlattice peak is narrow with a half-width-at-half-maximum (HWHM) ~ 0.001 in reciprocal lattice units and doubles the unit cell along the crystal c-axis.

Since $Q_A^{\perp} = (0, 0, L=$half-integer) overlaps with the A-type AFM peaks, it is necessary to prove that the observed peak at $Q_A^{\perp}$ is not due to the magnetic cross-section of x-ray scattering. For this purpose, we determine the temperature dependent superlattice peaks at $Q_L = (0, 0.5, 2.5)$, $Q_M^{//} = (0, 0.5, 3)$ and $Q_A^{\perp} = (0, 0, 2.5)$ and (0, 0, 4.5). Fig. 2a, c, e show θ−2θ scans below (90 K) and above (116 K) $T_{CDW}$. Fig. 2b, d, f show the full temperature dependent peak intensities and peak widths across the $T_{CDW}$. The same onset temperature for all four wave-vectors proves that $Q_A^{\perp}$ peaks correspond to charge superlattice along the c-axis. Since the x-ray scattering amplitude at $\mathbf{Q} = (0, 0, L)$ probes lattice distortions along the c-axis, the superlattice peaks at $Q_A^{\perp}$ establish an out-of-phase lattice distortions along the c-axis between adjacent FeGe layers at $T_{CDW}$. Insets of Fig. 2b, d show the hysteresis-scans at $Q_L$ and $Q_M^{//}$ near $T_{CDW}$. The small hysteresis temperature, $\Delta T \sim 0.5$ K, indicates that the transition at $T_{CDW}$ is a weak first-order transition[20]. We note that we do not observe the temperature dependent hysteresis at $Q_A^{\perp}$, possibly due to its relatively weak peak intensity near $T_{CDW}$ or an even smaller hysteresis temperature, $\Delta T$. Given both $Q_A^{\perp}$ and $Q_M^{//}$ are present in FeGe, the 2 × 2 × 2 superstructure peaks at $Q_L$ should be considered as a superposition of $Q_A^{\perp}$ and the three-equivalent $Q_M^{//}$. As we continue to discuss below, an important consequence of the $Q_A^{\perp}$-peak is that it distinguishes the 2 × 2 × 2 charge modulations in FeGe and AV₃Sb₅, pointing to different electronic and structural origins of the CDWs in these two kagome metals.

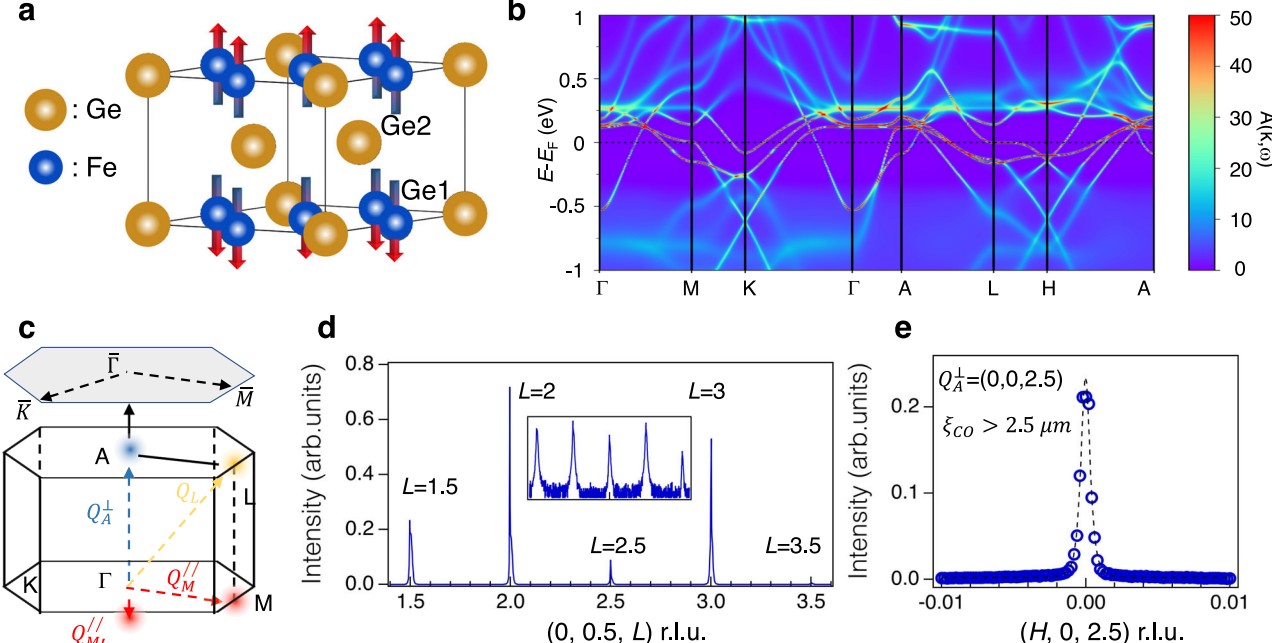

**Fig. 1 | Spin, charge, and lattice structures of FeGe. a** Crystal and magnetic structure of FeGe. **b** DFT + DMFT calculated electronic structure in the A-AFM phase with $U = 4.2$ eV and $J_H = 0.88$ eV showing van Hove singularities near the Fermi level, consistent with experiment[24]. **c** High symmetry points and directions in the non-magnetic bulk and surface (grey hexagon) Brillouin zone. Since the magnetic unit cell doubles the non-magnetic unit cell along the Γ-A direction, the magnetic Brillouin zone is half of the non-magnetic Brillouin zone along the Γ-A

direction (green dashed lines). $Q_A^{\perp}$ and $Q_M^{//}$ are corresponding to the charge-dimer and van Hove singularity nesting wavevectors, respectively. As discussed in the main text, $Q_L$ is naturally described as a superposition of $Q_A^{\perp}$ and $Q_M^{//}$. **d** L-scan along the [0, 0.5, L] direction. The inset shows the same intensity in log-scale. **e** H-scan at charge-dimer wavevector (0, 0, 2.5). The dashed curve is a fitting of the peak using Lorenzian-squared function. Data shown in **d** and **e** were taken at 10 K.

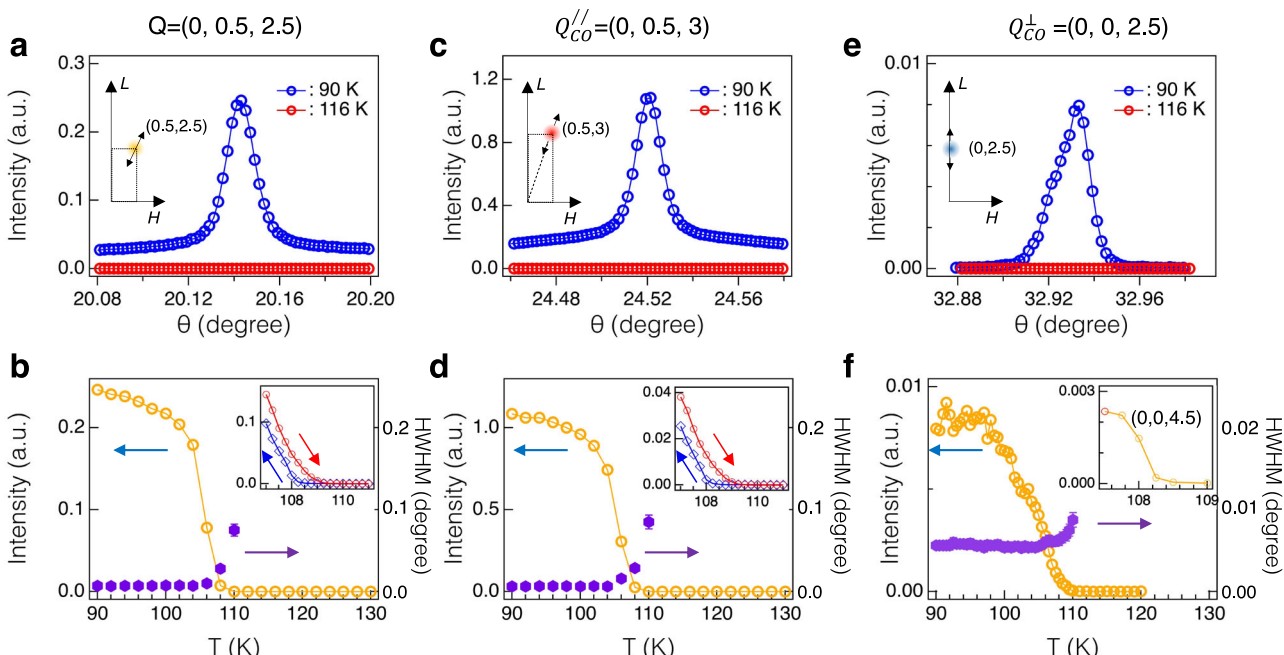

**Fig. 2 | Observation of charge dimerization superlattice peaks. a, c, e**, θ−2θ scans at $Q$ = (0, 0.5, 2.5), (0, 0.5, 3) and (0, 0, 2.5). Their corresponding trajectories in the momentum space are shown in the inset of **a, c, e**. Red and blue curves correspond to scans at $T$ = 116 and 90 K, respectively. The temperature dependent peak intensities at $Q_L$ = (0, 0.5, 2.5), $Q_M^{//}$ = (0, 0.5, 3) and $Q_A^{\perp}$ = (0, 0, 2.5) are shown in **b**, **d**, and **f**, respectively. Open and solid marks represent peak intensity and peak width, respectively. Insets of **b** and **d** show hysteresis scans of the peak intensity near $T_{CDW}$, uncovering a weak first order phase transition. The inset of **f** shows the temperature dependent peak intensity at another charge-dimer wavevector (0, 0, 4.5). All experimental data except the inset of **f** were collected at 4-ID, APS with photon energy $h\nu$=11 keV. The data shown in the inset of **f** is collected at 30-ID, APS with $h\nu$=23.71 keV.

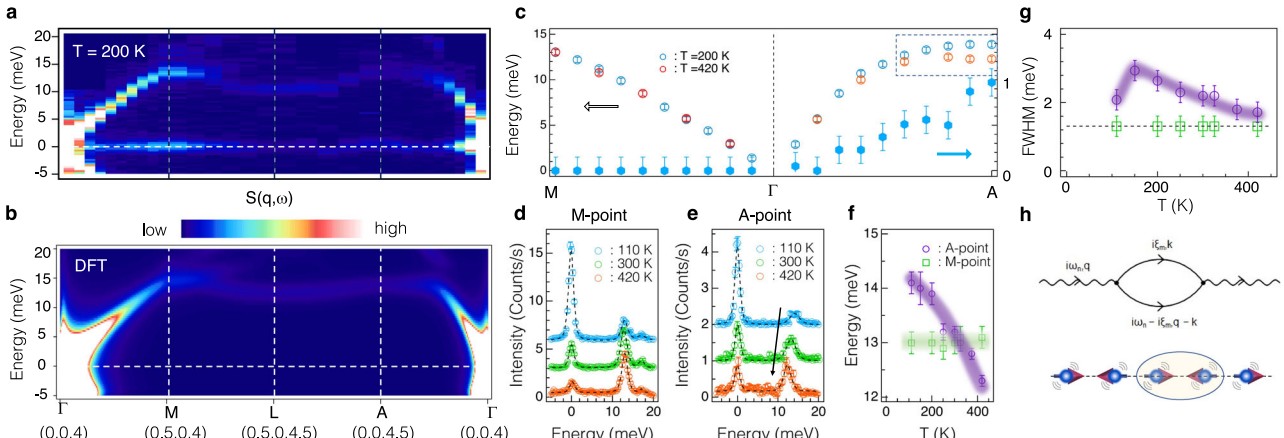

**Fig. 3 | Giant phonon anomalies near the charge dimerization wavevectors. a** and **b** experimental and DFT $S(Q, \omega)$ along the **Γ**(0, 0, 4)-**M**(0.5, 0, 4)-**L**(0.5, 0, 4.5)-**A**(0, 0, 4.5)-**Γ**(0, 0, 4) direction. As described in Methods, the IXS intensity is dominated by the lattice distortions along the crystal c-axis. The IXS data shown in **a** were collected at 200 K. **c** Extracted phonon dispersion along the **M-Γ-A** direction at 200 (cyan) and 420 K (orange). The dashed rectangle highlights the temperature dependent phonon energy renormalization near the A point. **d** and **e** temperature dependent IXS spectra at the M and A point. Dashed curves are fittings of the experimental data (see Methods). Blue, green and orange circles represent 110, 300 and 420 K, respectively. **f** Temperature dependence of the fitted phonon peak positions at the A (open purple circles) and M (open green squares) point. **g** Temperature dependence of the fitted phonon peak width at the A (open purple circles) and M (open green squares) point. Dashed line represents the instrumental energy resolution. **h** Dynamical spin-phonon coupling. Top panel shows the second-order Feynman diagram for the phonon self-energy. Dashed and solid lines represent the phonon and magnon Green's functions, respectively. $i\omega n$ and $i\xi_m$ are bosonic Matsubara frequencies. Bottom panel shows a schematic of the dynamical spin-phonon coupling in an effective 1-dimensional spin chain with A-AFM. The magnon-phonon scattering induces strong the phonon self-energy effects and yields a phonon-energy hardening and phonon-linewidth broadening near the charge dimerization A-point (see Supplementary Fig. 11). The vertical error bars shown in **c–f** represent 1-standard deviation from either Poissonian statistics or least-squares fitting. The vertical error bars shown in **g** represent the experimental step size that is about 3 times larger than the fitting error bars.

The observation of charge superlattice peaks at $Q_A^{\perp}$ on top of the A-AFM peaks naturally point to a spin-phonon interaction in FeGe. We thus turn to determine the phonon dynamical structure factor, $S(Q, \omega)$, using meV-resolution inelastic x-ray scattering (IXS). Fig. 3a, b show experimental and DFT calculated $S(Q, \omega)$ along the **Γ**(0, 0, 4)-**M**(0.5, 0, 4)-**L**(0.5, 0, 4.5)-**A**(0, 0, 4.5)-**Γ**(0, 0, 4) direction at 200 K. The overall agreements between IXS and DFT, including the phonon dispersion and intensity distribution, are good. However, we find that the phonon peak-widths near $Q_A^{\perp}$ are unusually broad, suggesting quasiparticle interactions[29–33]. To reveal more details of this phonon anomaly, we

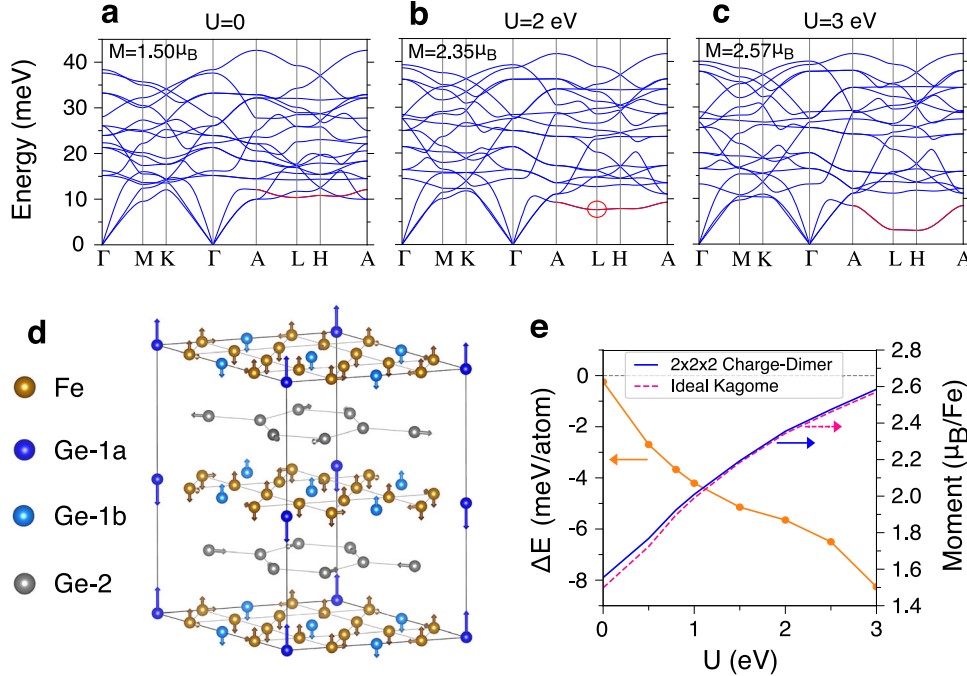

**Fig. 4 | Static spin-polarization-assisted 2 × 2 × 2 superstructure in FeGe. a–c,** The DFT + $U$ calculated phonon spectra of FeGe in the AFM phase as increasing Hubbard $U$. The phonon spectra are plotted with respect to the non-magnetic BZ of FeGe. The calculated ordered magnetic moments per Fe atom, M, are 1.50, 2.35 and 2.57 $\mu_B$ for $U=0$, 2 and 3 eV, respectively. The calculated M at $U=0$ eV is closer to the experimental value. The red curve corresponds to the experimentally observed phonon modes which show the most dramatic change as the spin-polarization is enhanced. The red circles highlight the $B_{1u}$ phonon mode at the L-point, which has the lowest energy along the **A-L-H-A** direction. The energy of the $B_{1u}$ mode and nearby phonon dispersion at $U=0$ matches the IXS determined dynamical structure factor shown in Fig. 3a. **d** The equal phase and amplitude superposition of the $B_{1u}$ modes at the three equivalent **L**-points yields a charge-dimerized 2 × 2 × 2 superstructure. The arrows indicate the movements of Fe and Ge atoms. In kagome layers, Fe and Ge atoms move out-of-plane to form dimers along c-axis. Ge-1a (blue) and Ge-1b (light blue) have out-of-phase vibrations. Ge-1a has much larger movement than Ge-1b. The honeycomb layers of Ge−2 (grey) atoms show in-plane Kekulé-type distortions. **e** Left y-axis shows the DFT + $U$ calculated energy difference between the charge-dimerized 2 × 2 × 2 superstructure and the ideal Kagome phase, $\Delta E = E_{Charge-Dimer} - E_{Kagome}$. Right y-axis shows the calculated ordered magnetic moment of the 2 × 2 × 2 superstructure (blue solid) and ideal Kagome (red dashed) phases, respectively. The magnetic moments are enhanced by 0.01 - 0.05 $\mu_B$/Fe by forming the 2 × 2 × 2 superstructure.

show in Fig. 3c the phonon band dispersion at 200 and 420 K (open circles) and the extracted phonon peak width at 200 K (solid circles) along the **M-Γ-A** direction. Interestingly, the phonon energy at the **A**-point shows over 10% hardening from 420 to 200 K that accompanies with a phonon linewidth broadening. Fig. 3d, e show representative temperature dependent IXS spectra at the **M** and **A** point, respectively. The full temperature dependence of the extracted peak positions is shown in Fig. 3f. We find that the phonon peaks at the **M** point remain temperature independent within experimental error, whereas the phonon mode at the **A** point shows a giant 14% hardening from 420 to 110 K. Fig. 3g summarizes the temperature dependent phonon width at the **A** and **M** point. The fitted phonon peak-width at the **A**-point is broad even above the $T_N$ with full-width-at-half-maximum (FWHM) ~ 2 meV. The width continuously increases until temperature hits the $T_{CDW}$. At $T = 150$ K, the fitted FWHM ~ 3 meV (corresponding to ~ 2 meV intrinsic phonon width after deconvolution), yielding a Damping ratio~7%. In stark contrast, the phonon peak width at the **M** and **L** (see Supplementary Fig. 4) point is resolution limited in the entire temperature range, consistent with the absence of longitudinal and transverse acoustic phonon energy anomaly at the **M** and **L** point.

The observed phonon hardening and broadening above the $T_{CDW}$ in FeGe are fundamentally different from the Kohn anomaly in electron-phonon coupled CDW systems and the emergent amplitude mode below the $T_{CDW}$[19, 29–31]. These phonon anomalies are, however, captured by the dynamical spin-phonon coupling picture shown in Fig. 3h. The second order Feynman diagram depicts a phonon with energy, $\omega_n$, and momentum, $q$, scatters into two magnons with ($\xi_m$, $k$) and ($\omega_n$-$\xi_m$, $q$-$k$). As we show in more details in the Supplementary

Discussion and Supplementary Fig. 11, this dynamical spin-phonon interaction yields strong phonon self-energy effect, including the phonon energy hardening and phonon linewidth broadening near the A-point, in agreement with experimental observations. Interestingly, similar phonon anomalies were observed in Kondo insulator FeSi[33] and spin-Peierls compound CuGeO$_3$[34,35], supporting a ubiquitous phonon hardening and broadening in spin-phonon coupled systems. The observation of superlattice peaks at $Q_A^\perp$ and the associated giant phonon anomalies constitute our main experimental observations.

Motivated by these experimental results, we perform DFT + $U$ calculations for the A-AFM phase of FeGe at the zero temperature to understand the interplay between static spin-polarizations and the lattice distortions in the CDW phase. Fig. 4a–c show the calculated phonon spectra of FeGe in the A-AFM phase as increasing Hubbard $U$. We find that the experimentally observed phonon modes shown in Fig. 3a exhibit the most dramatic change as increasing $U$ with an energy minimum at the **L** point for U < 2 eV. This observation indicates that stronger electronic correlations and spin-polarizations tend to induce a lattice instability in FeGe. Interestingly, this mode corresponds to atomic vibrations that are mainly composed of out-of-phase c-axis lattice distortions between adjacent Fe-Ge kagome layers, consistent with experimentally observed superlattice peaks at $Q_A^\perp$. To understand the nature of the 2 × 2 × 2 superstructure, we take the equal phase and amplitude superpositions of the experimentally observed phonon mode at the three equivalent **L**-points as shown in Fig. 4d. The arrows point the movement of Fe and Ge atoms. In kagome layers, Fe and Ge atoms move out-of-plane along the c-axis. The Ge-1 atoms are divided into out-of-phase Ge-1a (blue) and Ge-1b (light blue) groups, where Ge-

1a has a much larger atomic movement than that of Ge-1b, forming Ge-1 dimers along the c-axis. The honeycomb layers of Ge−2 atoms (grey) show in-plane Kekulé-type distortions[36,37]. Starting from this $2 \times 2 \times 2$ supercell that preserves the $P6/mmm$ space group, we relax the internal atomic positions. Fig. 4e shows the energy difference between the $2 \times 2 \times 2$ superstructure and the ideal kagome phase, $\Delta E = E_{Charge-Dimer} - E_{Kagome}$, which decreases as increasing $U$. Intriguingly, the $2 \times 2 \times 2$ superstructure is already an energetically favored phase at $U = 0$ and becomes even more robust with increasing $U$ accompanied by the increase in the static moment. These results suggest that the magnitude of the static spin-polarization is important to stabilize the $2 \times 2 \times 2$ superstructure with large c-axis lattice distortions in the Kagome plane. Furthermore, as we show in Fig. 4e, by forming this $2 \times 2 \times 2$ superstructure, the ordered magnetic moment is further enhanced by 0.01 ~ 0.05 $\mu_B$/Fe at $U = 0 ~ 3$ eV, consistent with the previous neutron scattering study[24]. It is important to point out that the experimentally determined static spin moment is more consistent with DFT + U calculations at $U = 0$ (Fig. 4a), therefor the static spin-moment induced phonon softening effect at elevated temperature will be neglectable and the dominated phonon anomaly is expected to arise from the dynamical magnon-phonon coupling as shown in Fig. 3.

Our experimental and numerical results support a spin-phonon coupling picture for the emergence of CDW in FeGe. Near $T_{CDW}$, the energy gain by forming a $2 \times 2 \times 2$ superstructure with enhanced static moment overcomes the energy cost of lattice distortions and gives rise to a weak first order phase transition[38]. The presence of large itinerant electrons allows additional energy gain by removing the high density-of-states near $E_F$[39]. We emphasize, however, that the A-AFM induced van Hove singularity near $E_F$ may only plays a minor role for the CDW in FeGe for the following reasons: first, the conventional electron-phonon coupling tends to favor lattice distortions parallel to the nesting vectors different from the experimental and DFT observations. Second, the strong temperature dependent phonon anomaly near $Q_A^{\perp}$ and the absence of phonon anomaly at $Q_M^{//}$ are incompatible with a nesting driven CDW picture.

## Methods

### Sample preparation and characterizations
Single crystals of B35-type FeGe were grown via chemical vapor transport method. Stoichiometric iron powders (99.99%) and germanium powders (99.999%) were mixed and sealed in an evacuated quartz tube with additional iodine as the transport agent. The quartz tube was then loaded into a two-zone horizontal furnace with a temperature gradient from 600 °C (source) to 550 °C (sink). After 12 days growth, FeGe single crystals with typical size $1.5 \times 1.5 \times 3$ mm$^3$ can be obtained in the middle of the quartz tube.

### Elastic X-ray scattering
The single crystal elastic X-ray diffraction was performed at the 4-ID-D beamline of the Advanced Photon Source (APS), Argonne National Laboratory (ANL). The incident photon energy was set to 11 keV, slightly below the Ge K-edge to reduce the fluorescence background. The X-rays higher harmonics were suppressed using a Si mirror and by detuning the Si (111) monochromator. Diffraction was measured using a vertical scattering plane geometry and horizontally polarized (σ) X-rays. The incident intensity was monitored by a He filled ion chamber, while diffraction was collected using a Si-drift energy dispersive detector with approximately 200 eV energy resolution. The sample temperature was controlled using a He closed cycle cryostat and oriented such that X-rays scattered from the (001) surface.

### meV-resolution inelastic X-ray scattering
The experiments were conducted at beam line 30-ID-C (HERIX) at APS, ANL[40]. The highly monochromatic x-ray beam of incident energy $E_i$ = 23.7 keV (λ = 0.5226 Å) was focused on the sample with a beam cross section of $\sim 35 \times 15$ μm$^2$ (horizontal × vertical). The overall energy resolution of the HERIX spectrometer was ΔE $\sim$ 1.5 meV (full width at half maximum). The measurements were performed in reflection geometry. Under this geometry, IXS is primarily sensitive the lattice distortions along the crystal c-axis. This geometry selectively enhances the unstable phonon modes predicted in the DFT calculations. Typical counting times were in the range of 120 to 240 seconds per point in the energy scans at constant momentum transfer **Q**. $H, K, L$ are defined in the hexagonal structure with a = b = 4.97 Å, c = 4.04 Å at the room temperature.

### Curve Fitting
The total energy resolution $\Delta E = 1.5$ meV is calibrated by fitting the elastic peak to a pseudo-voigt function:

$$R(\omega) = (1 - \alpha) \frac{I}{\sqrt{2\pi}\sigma} e^{-\frac{\omega}{2\sigma^2}} + \alpha \frac{I}{\pi} \frac{\Gamma}{\omega^2 + \Gamma^2} \qquad (1)$$

where the energy resolution is the FWHM.

IXS directly probes the phonon dynamical structure factor, $S(\mathbf{Q}, \omega)$. The IXS cross-section for solid angle $d\Omega$ and bandwidth $d\omega$ can be expressed as:

$$\frac{d^2\sigma}{d\Omega d\omega} = \frac{k_f}{k_i} r_0^2 |\vec{\epsilon}_i \cdot \vec{\epsilon}_f|^2 S(\mathbf{Q}, \omega) \qquad (2)$$

where **k** and $\boldsymbol{\epsilon}$ represent the scattering vector and x-ray polarization and $i$ and $f$ denote initial and final states. $r_O$ is the classical radius of the electron. In a typical measurement, the energy transfer $\omega$ is much smaller than the incident photon energy (23.71 keV in our study). Therefore, the term $\frac{k_f}{k_i} \sim 1$, and $\frac{d^2\sigma}{d\Omega d\omega} \propto S(\mathbf{Q}, \omega)$.

$S(\mathbf{Q}, \omega)$ is related to the imaginary part of the dynamical susceptibility, $\chi''(\mathbf{Q}, \omega)$, through the fluctuation-dissipation theorem:

$$S(\mathbf{Q}, \omega) = \frac{1}{\pi} \frac{1}{(1 - e^{\omega/k_B T})} \chi''(\mathbf{Q}, \omega) \qquad (3)$$

Where $\chi''(\mathbf{Q}, \omega)$ can be described by the damped harmonic oscillator form, which has antisymmetric Lorentzian lineshape:

$$\chi''(\mathbf{Q}, \omega) = \sum_i I_i \left[ \frac{\Gamma_i}{(\omega - \omega_{Q,i})^2 + \Gamma_i^2} - \frac{\Gamma_i}{(\omega + \omega_{Q,i})^2 + \Gamma_i^2} \right] \qquad (4)$$

here $i$ indexes the different phonon peaks.

The phonon peak can be extracted by fitting the IXS spectrum at constant-momentum transfer **Q**, using Eqs. (3) and (4). Due to the finite experimental resolution, the IXS intensity is a convolution of $S(\mathbf{Q}, \omega)$ and the instrumental resolution function, $R(\omega)$:

$$I(\mathbf{Q}, \omega) = S(\mathbf{Q}, \omega) \otimes R(\omega) \qquad (5)$$

Here $R(\omega)$ was determined by fitting of the elastic peak.

### ARPES experiment
The ARPES experiments are performed on single crystals FeGe. The samples are cleaved in situ in a vaccum better than $5 \times 10^{-11}$ torr. The experiment is performed at beam line 21-ID-1 at the NSLS-II. The measurements are taken with synchrotron light source and a Scienta-Omicron DA30 electron analyzer. The total energy resolution of the ARPES measurement is approximately 15 meV. The sample stage is maintained at $T = 30$ K throughout the experiment.

## DFT + U calculations

DFT + U calculations are performed using Vienna ab initio simulation package (VASP)[41]. The exchange-correlation potential is treated within the generalized gradient approximation (GGA) of the Perdew-Burke-Ernzerhof variety[42]. The simplified approach introduced by Dudarev et al. (LDAUTYPE = 2) is used[43]. We used experimental lattice parameters of FeGe and FeSn[24,44]. Phonon calculations are performed in the A-type AFM phase with a $2 \times 2 \times 1$ supercell (with respect to the AFM cell), using both the density-functional-perturbation theory (DFPT)[45] and frozen phonon approaches, combined with the *Phonopy* package[46]. The two approaches yield identical results. The internal atomic positions of the charge-dimerized $2 \times 2 \times 2$ superstructure is relaxed with the initial atomic distortions shown in Fig. 4d, until the force is less than 0.001 eV/Å for each atom. Integration for the Brillouin zone is done using a Γ-centered $8 \times 8 \times 10$ $k$-point grids for the $2 \times 2 \times 2$ supercell and the cutoff energy for plane-wave-basis is set to be 500 eV. Besides the $2 \times 2 \times 2$ lattice distortion ansatz, we have also employed other lattice distortion ansatz, including $1 \times 2 \times 2$, $\sqrt{3} \times \sqrt{3} \times 2$ and $\sqrt{5} \times \sqrt{5} \times 2$. All these ansata yield ground state energies higher than the $2 \times 2 \times 2$ superstructure and the original ideal Kagome structure.

## DFT + DMFT calculations

The fully charge self-consistent DFT + DMFT[47] calculations are performed in the A-type AFM phase using an open-source code of DFT +embedded DMFT developed by Haule et al., based on Wien2k package[48]. We choose a hybridization energy window from −10 eV to 10 eV with respect to the Fermi level. All the five 3d orbitals on an Fe site are considered as correlated ones, and a local Coulomb interaction Hamiltonian of Ising form is applied with varied Hubbard $U$ and Hund's coupling $J_H$ as shown in the main text. We use the continuous time quantum Monte Carlo[49] as the impurity solver and an "exact" double counting scheme by Haule[50,51]. To compute the spectral function, the electron self-energy on real frequency is obtained by the maximum entropy analytical continuation method. The SOC is not included in the DFT + DMFT calculations since the SOC strength of Fe-3d orbitals is small and will rarely change the electronic correlations. All the calculations are performed at $T = 80$ K.

## Electron-phonon vs spin-phonon driven CDW

From an energy point of view, the electron-phonon coupling driven CDW emphasizes the competing energy scales of charge condensation energy and lattice deformation energy, whereas the spin-phonon coupling highlights the magnetic energy gain by forming a CDW. To understand the spin-phonon coupling driven CDW, one can consider a simplified 1D Heisenberg model:

$$H = J \sum_{i=1}^{N} (1 + \Delta_i) \boldsymbol{S}_i \cdot \boldsymbol{S}_{i+1} + \frac{k}{2} \sum_{i}^{N} \Delta_i^2$$

Here $J$ is the antiferromagnetic exchange energy, $\boldsymbol{S}_i$ is the local spin. $\Delta_i = (-1)^i \delta, \delta \geq 0$ is the lattice distortion at bond $i$, connecting sites $i$ and $i+1$, and $k$ is the elastic constant. A CDW is energetically favored if the energy gain in the first magnetic term is greater than the energy cost of the second elastic term. This is rather static spin-phonon coupling. When the system is magnetically ordered, the energetics of this system is described by quasiparticles, *i.e.* magnons and phonons, thus the dynamical spin-phonon coupling becomes crucial. In the Supplementary Discussion (Section "*Phonon lifetime by two magnon excitations*"), we build a magnon-phonon coupling model on a 1D AFM Heisenberg chain. One of the consequences of such dynamical spin-phonon coupling appears as the phonon lifetime, which allows the direct comparison between experimental data and a theoretical prediction.

## Data availability

The data that support the findings of this study are available from the corresponding author upon request.

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

## Acknowledgements

We thank Matthew Brahlek, Pengcheng Dai, Jiangping Hu, H. C. Lei, Brain Sales, Jiaqiang Yan, Binghai Yan, Ming Yi, Zhida Song, and Jianzhou Zhao for stimulating discussions. This research was supported by the U.S. Department of Energy, Office of Science, Basic Energy Sciences, Materials Sciences and Engineering Division (x-ray and ARPES measurement and model analysis). This research used resources (beamline 4ID and 30ID) of the Advanced Photon Source, a U.S. DOE Office of Science User Facility operated for the DOE Office of Science by Argonne National Laboratory under Contract No. DE-AC02-06CH11357. ARPES measurements used resources at 21-ID-1 beamlines of the National Synchrotron Light Source II, a US Department of Energy Office of Science User Facility operated for the DOE Office of Science by Brookhaven National Laboratory under contract no. DE-SC0012704. T. T. Z. and S. M. acknowledge support from Tokodai Institute for Element Strategy (TIES) funded by MEXT Elements Strategy Initiative to Form Core Research Center Grants No. JPMXP0112101001, JP18J23289, JP18H03678, and JP22H00108. J.X.Y. acknowledges startup funding from the Southern University of Science and Technology. X.L.W. and A.F.W. acknowledge the support of the National Natural Science Foundation of China (Grant No. 12004056). T.T.Z. also acknowledges the support of the Japan Society for the Promotion of Science (JSPS), KAKENHI Grant No. JP21K13865. Y.L.W. acknowledges the support of the National Natural Science Foundation of China (No. 12174365). The DFT and DFT + DMFT calculations were performed on ThianHe-1A, the National Supercomputer Center in Tianjin, China. R.T. acknowledges the support from The São Paulo Research Foundation, FAPESP (Grant No. 2021/11170-0).

## Author contributions

H.M., A.F.W., Y.L.W. and S.O. conceived the project. H.M., H.X.L, G.F., J.X.Y. and R.S. performed the elastic X-ray scattering measurement. H.M., H.X.L., G.F. and A.S. performed the inelastic X-ray scattering. H.M., H.X.L., H.N.L., T.Y., E.V. performed the ARPES measurement. X.L.W. and A.F.W. grew the single crystals of FeGe. T.T.Z. and S.M. calculated the phonon dynamical structure factor. Y.L.W. performed DFT+U and DFT +DMFT calculations for FeGe and FeSn. L.X.F., K.J. and S.O. performed the 1D model Analysis. H.M., T.T.Z., S.O. and Y.L.W. wrote the paper with inputs from all co-authors.

## Competing interests

The authors declare no competing interests.
