## [Peer Review File · Nature Communications]

REVIEWER COMMENTS

Reviewer #2 (Remarks to the Author):

H. Miao et al. report the origin of the charge density wave in FeGe is driven by the spin-phonon coupling, different from the previous arguments that the CDW is Fermi surface nesting or electron-phonon coupling driven. The authors combine a complete elastic and inelastic X-ray scattering, and theoretical analysis to demonstrate the role of dynamical spin-phonon coupling. In my opinion, the data in this work are solid, the interpretation is reasonable, and the conclusion is creative. I am glad to recommend this work for publication in Nature Communications.

I do have some suggestions and hope the authors could consider in a revised version.

1. The authors should clarify the differences and connections between the electron-phonon coupling, the static spin/magnon-phonon coupling, and the dynamic spin/magnon-phonon coupling. Because these concepts were repeated in the manuscript several times and they are important for readers to understand the main point of this work.
2. In the dynamic spin-phonon coupling scenario, why is the temperature of CDW much lower than the AFM ordering temperature?
3. The hardening and broadening effects of the phonon appear at the A-point. Why is the A-point special?
4. In Fig. 1a, the arrows representing the moments of Fe are similar to the vibration mode of the ions in Fig. 4d. They are a little misleading. Arrows going through the Fe ions will be better.
5. The various colors against temperature on the plots in Fig. 2b, 2d, and 2f are not necessary. The data are simply temperature dependent. The color in the inset is red for 116 K, inconsistent with the various color.
6. The symbols "A-point" in Fig. 3d and "M-point" in Fig. 3e are required.
7. The FWHMs in Fig. 3g should be obtained by convoluting the instrumental resolution. It should be clarified. Because the resolutions at the M point are close to zero.

Reviewer #3 (Remarks to the Author):

The revised manuscript by Miao et al have clarified their notation of the CDW order, improved the experimental data which clarifies the phonon anomaly at the relevant high symmetry momenta, and provided more complete numerical calculations to support the claimed mechanism of CDW in FeGe. I appreciate the efforts made the authors that significantly improved strength of the statements made in the paper. Most of my questions have been addressed in the revised manuscript. Given the interest of this subject, after the following questions/points are properly addressed, I would recommend publication of the MS to Nature Communications.

1) The authors considered B_{1u} phonons in the DFT calculation. I want to clarify, is the B_{1u} phonon identified in the X-ray study or is hypothetical? What is exactly found in the X-ray study? I may missed it in the text, but I didn't find the explicit discussion.

2) To clarify the terminology, by CDW, does the DFT calculation see charge density modulation, or it is just lattice distortion? In cases, people mix the terminologies, but I think it should be clarified in the text.

3) I am not an expert on DFT, but it seems to me the DFT result is biased in this treatment because the authors use a lattice distortion ansatz in the calculation and then variationally lower the energy? I am concerned with it because the ΔE for small U is close to zero.

We thank all the reviewers for their time and efforts in reviewing our manuscript. Below we copied Reviewer's comments *in black* and replied to these comments *in blue*.

Reviewer #2:

H. Miao et al. report the origin of the charge density wave in FeGe is driven by the spin-phonon coupling, different from the previous arguments that the CDW is Fermi surface nesting or electron-phonon coupling driven. The authors combine a complete elastic and inelastic X-ray scattering, and theoretical analysis to demonstrate the role of dynamical spin-phonon coupling. In my opinion, the data in this work are solid, the interpretation is reasonable, and the conclusion is creative. I am glad to recommend this work for publication in Nature Communications.

Reply: We thank Reviewer #2 for his/her positive evaluation of our work.

I do have some suggestions and hope the authors could consider in a revised version.

1. The authors should clarify the differences and connections between the electron-phonon coupling, the static spin/magnon-phonon coupling, and the dynamic spin/magnon-phonon coupling. Because these concepts were repeated in the manuscript several times and they are important for readers to understand the main point of this work.

Reply: We appreciate this constructive suggestion. Quasiparticles in materials are subject of many body interactions, including electron-phonon coupling and spin-phonon coupling etc. Since CDW must involve lattice distortions, phonons are always critical for the emergence of CDW.

From an energy point of view, the electron-phonon coupling driven CDW emphasizes the competing energy scales of charge condensation energy and lattice deformation energy, whereas the spin-phonon coupling highlights the magnetic energy gain by forming a CDW. To understand the spin-phonon coupling driven CDW, one can consider a simplified 1D Heisenberg model:

$$H = J \sum_{i=1}^N (1 + \Delta_i) \mathbf{S}_i \cdot \mathbf{S}_{i+1} + \frac{k}{2} \sum_i^N \Delta_i^2$$

Here J is the antiferromagnetic exchange energy, \mathbf{S}_i is the local spin. $\Delta_i = (-1)^i \delta$, $\delta \geq 0$ is the lattice distortion at bond i , connecting sites i and $i+1$, and k is the elastic constant. A CDW is energetically favored if the energy gain in the first magnetic term is greater than the energy cost of the second elastic term. This is rather static spin-phonon coupling. When the system is magnetically ordered, the energetics of this system is described by quasiparticles, *i.e.* magnons and phonons, thus the dynamical spin-phonon coupling becomes crucial. In the supplementary materials (Section "Phonon lifetime by two magnon excitations"), we build a magnon-phonon coupling model on a 1D AFM Heisenberg chain. One of the consequences of such dynamical spin-phonon coupling appears as the phonon lifetime, which allows the direct comparison between experimental data and a theoretical prediction.

Following Reviewer #2's suggestion, we added a new section in the method part to explain the difference between electron-phonon and spin-phonon coupling driven CDWs.

2. In the dynamic spin-phonon coupling scenario, why is the temperature of CDW much lower than the AFM ordering temperature?

Reply: This is because magnons are well-defined only in the magnetic ordered phase. In FeGe, the static magnetic moment S_i increases with decreasing temperature, hence the magnon excitation energy is increased. Since the dynamical spin-phonon coupling requires the overlap of phonon dispersions and magnon dispersions, the temperature of CDW should be much lower than the AFM ordering temperature.

3. The hardening and broadening effects of the phonon appear at the A-point. Why is the A-point special?

Reply: Because of the layered-type AFM ordering in FeGe, the low-energy magnon excitation along the z direction appears along the Γ -A direction. Such magnon excitation energy is maximized at the middle point between Γ and A, allowing the strong coupling between magnons and phonons around the A point because two-magnon creation with zero total spin is allowed. This picture is consistent with the charge dimerization along the c-axis.

4. In Fig. 1a, the arrows representing the moments of Fe are similar to the vibration mode of the ions in Fig. 4d. They are a little misleading. Arrows going through the Fe ions will be better.

Reply: We modified Fig. 1a following Reviewer #2's suggestion.

5. The various colors against temperature on the plots in Fig. 2b, 2d, and 2f are not necessary. The data are simply temperature dependent. The color in the inset is red for 116 K, inconsistent with the various color.

Reply: The colors are removed from Fig. 2b, 2d and 2f.

6. The symbols "A-point" in Fig. 3d and "M-point" in Fig. 3e are required.

Reply: We added high-symmetry point notions in Fig. 3d and 3e.

7. The FWHMs in Fig. 3g should be obtained by convoluting the instrumental resolution. It should be clarified. Because the resolutions at the M point are close to zero.

Reply: Following Reviewer #2's suggestion, we updated Fig. 3g by showing the FWHM that includes the instrumental resolution. We also updated the text to reflect these changes.

Reviewer #3 (Remarks to the Author):

The revised manuscript by Miao et al have clarified their notation of the CDW order, improved the experimental data which clarifies the phonon anomaly at the relevant high symmetry momenta, and provided more complete numerical calculations to support the claimed mechanism

of CDW in FeGe. I appreciate the efforts made the authors that significantly improved strength of the statements made in the paper. Most of my questions have been addressed in the revised manuscript. Given the interest of this subject, after the following questions/points are properly addressed, I would recommend publication of the MS to Nature Communications.

Reply: We thank Reviewer #3 for his/her support on the publication of our results in *Nature Communications*. We are glad to see that most comments from Reviewer #3 were properly addressed.

1) The authors considered B_{1u} phonons in the DFT calculation. I want to clarify, is the B_{1u} phonon identified in the X-ray study or is hypothetical? What is exactly found in the X-ray study? I may missed it in the text, but I didn't find the explicit discussion.

Reply: We apologize for the ambiguity. The B_{1u} phonon mode at the L-point has been identified in the X-ray study (Fig. 3a and Fig. S4 in the supplementary materials). These phonon energy matches well with the DFT calculation with $U=0$ (Fig. 4a). As we pointed out in the manuscript “the experimentally determined static spin moment is more consistent with DFT+ U calculations at $U=0$ ”.

In the caption of figure 4, we added: “The energy of the B_{1u} mode and nearby phonon dispersion at $U=0$ matches the IXS determined dynamical structure factor shown in Fig. 3a”

2) To clarify the terminology, by CDW, does the DFT calculation see charge density modulation, or it is just lattice distortion? In cases, people mix the terminologies, but I think it should be clarified in the text.

Reply: Yes, in addition to the lattice distortion, a $2 \times 2 \times 2$ charge density modulation has also been observed in our DFT calculation. As shown in Fig.R1, the DFT-calculated charge density map of the CDW ground state clearly shows a $2 \times 2 \times 2$ charge density modulation. We added Fig. R1 in the revised supplementary materials (Fig. S8).

Fig. R1: The DFT-calculated charge density map of the CDW ground state.

3) I am not an expert on DFT, but it seems to me the DFT result is biased in this treatment

because the authors use a lattice distortion ansatz in the calculation and then variationally lower the energy? I am concerned with it because the ΔE for small U is close to zero.

Reply: The lattice distortion ansatz, *i.e.*, the initial guess of the atomic distortions in the structural optimization by DFT, is guided by the DFT calculated phonon spectra (Fig. 4) and the experimentally observed $2 \times 2 \times 2$ superstructure. But we have also employed other lattice distortion ansatz, including (i) $1 \times 2 \times 2$ superstructure, corresponding to any one of the three B_{1u} mode at L-point; (ii) $\sqrt{3} \times \sqrt{3} \times 2$ superstructure corresponding to the B_{1u} phonon mode at H-point; and (iii) a $\sqrt{5} \times \sqrt{5} \times 2$ superstructure. All these ansatz yield ground state energies higher than the $2 \times 2 \times 2$ superstructure and the original ideal Kagome structure. This indicates that the $2 \times 2 \times 2$ superstructure optimized by DFT is indeed the true ground state even though ΔE for small U is close to zero. This small energy difference at $U=0$ may be consistent with experimental observation where both the CDW transition temperature and correlation length can be reduced significantly by post annealing process (arXiv: 2308.01291).

We have added this discussion in the revised version Methods and Supplementary Materials (Caption of Fig. S5).